# Risk and Prognostic Factors in Very Old Patients with Sepsis Secondary to Community-Acquired Pneumonia

**DOI:** 10.3390/jcm8070961

**Published:** 2019-07-02

**Authors:** Catia Cillóniz, Cristina Dominedò, Antonella Ielpo, Miquel Ferrer, Albert Gabarrús, Denise Battaglini, Jesús Bermejo-Martin, Andrea Meli, Carolina García-Vidal, Adamanthia Liapikou, Mervyn Singer, Antoni Torres

**Affiliations:** 1Department of Pneumology, Hospital Clinic of Barcelona, 08036 Barcelona, Spain; 2August Pi i Sunyer Biomedical Research Institute–IDIBAPS, University of Barcelona, 08036 Barcelona, Spain; 3Biomedical Research Networking Centres in Respiratory Diseases (Ciberes), 28029 Madrid, Spain; 4Department of Anesthesiology and Intensive Care Medicine, Fondazione Policlinico Universitario A. Gemelli, Università Cattolica del Sacro Cuore, 00168 Rome, Italy; 5Departments of Medicine and Surgery, Respiratory Disease and Lung Function Unit, University of Parma, 43121 Parma, Italy; 6Department of Surgical Sciences and Integrated Diagnostic, Policlinico San Martino, University of Genova, 16126 Genova, Italy; 7Group for Biomedical Research in Sepsis (Bio Sepsis), Hospital Clínico Universitario de Valladolid/IECSCYL, Av. Ramón y Cajal, 3, 47003 Valladolid, Spain; 8Department of Anesthesia and Intensive Care, University of Milan, 20122 Milan, Italy; 9Infectious Diseases Department, Hospital Clinic of Barcelona, 08036 Barcelona, Spain; 10Respiratory Department, Sotiria Chest Diseases Hospital, Mesogion 152, 11527 Athens, Greece; 11Bloomsbury Institute of Intensive Care Medicine, Division of Medicine, University College London, Cruciform Building, Gower St, London WC1E 6BT, UK

**Keywords:** sepsis, community-acquired pneumonia, very old, pneumonia

## Abstract

**Background**: Little is known about risk and prognostic factors in very old patients developing sepsis secondary to community-acquired pneumonia (CAP). **Methods**: We conducted a retrospective observational study of data prospectively collected at the Hospital Clinic of Barcelona over a 13-year period. Consecutive patients hospitalized with CAP were included if they were very old (≥80 years) and divided into those with and without sepsis for comparison. Sepsis was diagnosed based on the Sepsis-3 criteria. The main clinical outcome was 30-day mortality. **Results**: Among the 4219 patients hospitalized with CAP during the study period, 1238 (29%) were very old. The prevalence of sepsis in this age group was 71%. Male sex, chronic renal disease, and diabetes mellitus were independent risk factors for sepsis, while antibiotic therapy before admission was independently associated with a lower risk of sepsis. Thirty-day and intensive care unit (ICU) mortality did not differ between patients with and without sepsis. In CAP-sepsis group, chronic renal disease and neurological disease were independent risk factors for 30-day mortality. **Conclusion**: In very old patients hospitalized with CAP, in-hospital and 1-year mortality rates were increased if they developed sepsis. Antibiotic therapy before hospital admission was associated with a lower risk of sepsis.

## 1. Introduction

Community-acquired pneumonia (CAP) is a major health problem in the elderly, being associated with high rates of readmission, morbidity, and mortality [1,2,3]. In the elderly, CAP worsens pre-existing comorbidities, and these interact to impact upon the clinical evolution of infection. The number of people aged ≥80 years has increased rapidly in Europe [4]. Rates of intensive care unit (ICU) admission have also increased for patients in this age group [5,6] and now account for approximately 10% of all ICU admissions in Europe [7,8].

Sepsis is a frequent complication of CAP [9]. However, this is a multifactorial process that requires staging to ensure treatments that target the needs of the individual patient [10]. A recent multicenter study found that age was an independent risk factor for mortality in sepsis, with prompt therapy provided within the first six hours of resuscitation being associated with a mortality decrease in very elderly (≥80 years) patients [11]. In a Spanish multicenter cohort study of 4070 patients hospitalized with CAP, 38% presented with severe sepsis (i.e., organ dysfunction) [9], for which age ≥65 years, chronic obstructive pulmonary disease, renal disease, and alcohol abuse were independent risk factors. Conversely, prior antibiotic therapy and diabetes mellitus were protective. Despite the increased risk of infectious disease due to age-related vulnerability, there is a scarcity of data on sepsis in very old patients with CAP [12].

We thus aimed to determine risk and prognosis factors associated with sepsis, as defined according to the Sepsis-3 criteria [10], in very old (≥80 years) patients hospitalized with CAP in comparison to their peers admitted without evidence of sepsis, and to investigate clinical outcomes.

## 2. Methods

### 2.1. Study Design and Patients

This was a retrospective observational study of prospectively collected data from the Hospital Clinic of Barcelona, Spain. We enrolled all consecutive adult patients with a diagnosis of CAP admitted to hospital via the emergency department between January 2005 and December 2017. We included patients from nursing homes, as we previously demonstrated that the microbial etiology in this population is similar to that of CAP arising in people living in their own homes [13]. Among all patients with CAP, we selected those aged ≥80 years and compared those with and without sepsis. We excluded patients with severe immunosuppression, such as human immunodeficiency viral infection, active solid or hematologic malignancy treated with chemotherapy, oral corticosteroid treatment with at least 20 mg prednisone (or equivalent) per day for at least two weeks, and treatment with other immunosuppressive drugs. We also excluded those with active tuberculosis or a confirmed alternative diagnosis.

For publication purposes, the study was approved by the ethics committee of our institution (Comité Ètic d’Investigació Clínica, register: 2009/5451). The need for written informed consent was waived because of the non-interventional study design.

### 2.2. Definitions

Very old patients were defined as those aged 80 years and above [5]. Pneumonia (CAP) was defined as a new pulmonary infiltrate on chest X-ray during hospital admission with symptoms and signs of a lower respiratory tract infection. Severe CAP was diagnosed by the presence of at least one major or three minor criteria of the Infectious Disease Society of America/American Thoracic Society (IDSA/ATS) guidelines [14]. Polymicrobial pneumonia was defined as pneumonia due to more than one pathogen.

Prior antibiotic treatment was defined as antibiotic intake during the week before hospital admission. The appropriateness of empiric antibiotic treatment was determined according to multidisciplinary guidelines for the management of CAP [15].

Sepsis was defined according to the criteria of the Third International Consensus Definitions for Sepsis and Septic Shock (Sepsis-3) as the presence of pneumonia and an increase ≥2 points in the Sequential Organ Failure Assessment (SOFA) score [10]. The presence of sepsis was evaluated at hospital admission when diagnosing CAP. The presence of an acute respiratory distress syndrome (ARDS) was evaluated within the first 24 h of hospital admission based on the Berlin definition [16].

### 2.3. Data Collection, Evaluation, and Microbiological Diagnosis

Demographic variables, comorbidities, and physiologic parameters were collected in the Emergency Department within 24 h of admission. The Pneumonia Severity Index (PSI), CURB-65 score (i.e., confusion, urea nitrogen, respiratory rate, blood pressure, and age ≥65 years,) and SOFA score at admission were calculated [17,18,19]. During hospitalization, we recorded whether the patients had specific complications, including multilobar infiltration, pleural effusions, ARDS [16], septic shock [20], and acute renal failure [21]. Further details are reported elsewhere [22]. All surviving patients were visited or contacted by telephone 30 days after discharge, and the hospital records and the database of Catalunya Health Department reviewed at one year. The criteria for etiological diagnosis are described elsewhere [22,23] and in Appendix A.

### 2.4. Outcomes

The primary outcome was 30-day mortality. Secondary outcomes were in-hospital mortality, 1-year mortality, ICU admission, ICU mortality, and need for mechanical ventilation.

### 2.5. Statistical Analysis

Numbers and percentages are reported for categorical variables, medians and interquartile ranges (IQRs) for continuous variables with non-normal distributions, and means and standard deviations (SDs) for those continuous variables with a normal distribution. Categorical variables were compared using the *χ*^2^ test or Fisher exact test, whereas continuous variables were compared using the *t*-test or nonparametric Mann–Whitney *U* test. Trends in associated factors were analyzed using the Mantel–Hansel test or linear regression for categorical and continuous variables, respectively.

Logistic regression analyses [24] were used to examine associations between sepsis and risk factors. In the first step, each risk factor was tested individually. In the second step, all risk factors showing an association in the univariate model (*p* < 0.10) were added to multivariable models. Finally, a backward stepwise selection (*p*_in_ < 0.05, *p*_out_ > 0.10) was used to determine factors associated with sepsis [25]. The Hosmer–Lemeshow goodness-of-fit test was performed to assess the overall fit of the final model.

Generalized linear model analyses [26] were then performed to determine the influence of the identified risk factors on 30-day mortality in patients with sepsis. Models were defined by a binomial probability distribution and a logit link function, using inverse probability of treatment weights [27] to account for biases due to observed confounders. First, each risk factor was tested individually. Second, a propensity score was developed for patients with sepsis, irrespective of the outcome, by using multivariate logistic regression to predict the influence of 15 predetermined variables on the presence of sepsis. Variables were chosen for inclusion in the propensity score calculation based on previously described methods [28]. Variables associated with sepsis and outcome were included (e.g., gender, smoking status, alcohol consumption, influenza vaccine, pneumococcal vaccine, previous inhaled corticosteroids, previous systemic corticosteroids, previous antibiotic within a week of admission, chronic pulmonary disease, chronic cardiovascular disease, chronic renal disease, chronic liver disease, chronic neurological disease, diabetes mellitus, and nursing home residency). The propensity score then informed the weight in the inverse probability of treatment weighting. Finally, the weighting and year of admission were incorporated into the multivariable (weighted) logistic regression model of 30-day mortality. Risk factors that showed an association in the univariate analyses (*p* < 0.10) were identified after a stepwise backward elimination procedure in which non-significant variables were dropped until no further improvement of the Akaike’s Information Criterion was achieved [29]. Multi-collinearity was checked by calculating the variance inflation factor. Odds ratios (ORs) and their 95% confidence intervals (CIs) were then calculated. The area under the receiver operating characteristic (ROC) curve was calculated for the ability to predict sepsis and 30-day mortality, using significant variables derived from the respective logistic regression models. To avoid confounding, patients with do-not-resuscitate (DNR) orders were not included in the evaluation of mortality outcome [30].

Possible overfitting and instability of the selection variables in the final model were measured by internal validation using ordinary nonparametric bootstrapping with 1000 bootstrap samples and bias-corrected accelerated 95% CIs [31]. A multiple imputation method [32] was sued for missing data in the multivariable analyses.

Unless otherwise stated, the level of significance was two-tailed and set at 0.05. All analyses were performed using IBM SPSS Version 23.0 (IBM Corp., Armonk, New York, USA).

## 3. Results

### 3.1. Overall Population

In total, 4219 patients with CAP were admitted to our hospital during the study period, with 2981 excluded from the final analysis (Figure 1). The study population therefore comprised 1238 very old patients with CAP. Additional data regarding the differences between old (age 65–79 years) and very old patients (≥80 years) are shown in Appendix A, between patients from nursing homes and their own homes in Appendix A, between those admitted and not admitted to ICU in Appendix A, and between those with and without septic shock in Appendix A. The rates of sepsis by three-year admission epochs are shown in Appendix A

### 3.2. Comparison of Very Old Patients With and Without Sepsis

Of the 1238 very old patients with CAP, 880 (71%) had sepsis. Table 1 summarizes the demographic and clinical characteristics of patients by the presence or absence of sepsis.

Compared to the no-sepsis group, the sepsis group was more likely to be younger, male, have diabetes mellitus and chronic renal disease, and have had pneumonia in the last year. They were less likely to have received previous antibiotic therapy. At admission, an uncommon presentation of pneumonia was frequently observed in patients with and without sepsis (Table 1). However, a lower percentage of pleuritic pain and a higher rate of altered mental status were observed in septic patients. Moreover, the sepsis group had lower lymphocyte counts at admission. Notably, 61 patients (7%) presented with septic shock at admission.

A higher percentage of patients in the sepsis group was classified as PSI risk classes IV–V, and more experienced a higher rate of severe CAP compared with the no-sepsis group.

An etiologic diagnosis was achieved more often in the sepsis group (34% vs. 27%; *p* = 0.01). Appendix A shows no significant difference between the two groups in the distribution of identified pathogens. Seven cases (1%) of multidrug-resistant (MDR) pathogens were identified in the sepsis group (5 *P. aeruginosa*; 1 *P. aeruginosa*, *S. aureus*, and *S. pneumoniae*; and 1 caused by *P. aeruginosa* and influenza virus A) compared with one case (0.2%) of an MDR pathogen (*P. aeruginosa*) in the no-sepsis group.

Data were available for empiric antibiotic treatment in 1224 (99%) patients. The most frequent regimens were ß-lactam plus macrolide (29%) or fluoroquinolone (24%). The sepsis group (26%) more often received fluoroquinolones plus ß-lactams than the no sepsis group (20%; *p* = 0.021). Inappropriate empiric treatment rates were comparable (3% and 4%; *p* = 0.57).

### 3.3. Risk Factors for Sepsis

The univariate logistic regression analyses identified several variables that were significantly associated with the development of sepsis in very old patients with CAP (Table 2).

In multivariable analysis, male sex, diabetes mellitus, and chronic renal disease were risk factors for sepsis, while the only protective factor was previous antibiotic therapy. The area under the ROC curve was 0.62 (95% CI 0.59 to 0.65) for the predictive model of sepsis. Internal validation of the final model was conducted using a bootstrapping procedure with 1000 samples and demonstrated robust results: all variables remained significant with small 95% CIs around the original coefficients.

### 3.4. Outcomes of Very Old Septic and Non-Septic Patients

Thirty-day and ICU mortality did not differ between the sepsis and no-sepsis groups (Table 3).

However, in-hospital mortality and one-year mortality were significantly higher in the sepsis group who more frequently needed ICU admission, mechanical ventilation, and longer length of hospital stays.

### 3.5. Factors Associated with 30-Day Mortality in Patients with Sepsis

For analysis of factors associated with 30-day mortality, we excluded the 61 patients with septic shock as previous studies have shown that septic shock was the main risk factor for mortality in patients with severe CAP [33,34]. We also excluded 207 patients from the sepsis group who had DNR orders and 68 with missing data regarding a DNR order (Table 4).

In the propensity-adjusted multivariable analysis, chronic renal disease and neurologic disease were independently associated with increased 30-day mortality, while diabetes mellitus was independently associated with decreased 30-day mortality. The area under the ROC curve was 0.71 (95% CI 0.65–0.78) for the multivariable model of 30-day mortality. All variables remained significant after bootstrapping, with small 95% CIs around the original coefficients.

## 4. Discussion

To the best of our knowledge, this is the first study to report on sepsis, as defined using the Sepsis-3 criteria [10], in very old patients with CAP, including the clinical characteristics of these patients and risk factors for sepsis in this age group. There are four main findings of our study. First, 71% of very old patients hospitalized with CAP presented with sepsis and 7% presented with septic shock according to the Sepsis-3 definitions. Second, male sex, chronic renal disease, and diabetes mellitus were independent risk factors for presenting with sepsis, whereas antibiotic therapy in the week prior was a protective factor. Third, both in-hospital and one-year mortality were higher in patients with sepsis than those without. Fourth, chronic renal and neurologic diseases were independently associated with increased 30-day mortality, whereas diabetes mellitus was associated with lower 30-day mortality.

Aging is associated with an increased prevalence of chronic conditions. Age-related changes in the immune system increase susceptibility to infectious diseases and sepsis. In this study, sepsis affected 71% of very old patients with CAP, 7% presented with septic shock, and 11% of those with sepsis required ICU admission. The prevalence of sepsis in our patients with CAP was almost double that previously reported by Montull et al. (38% of severe sepsis in patients with CAP) [9]. However, we used the current Sepsis-3 definition with a change in SOFA score ≥2; we included nursing home residents and focused on very old patients. In our study, we also observed an uncommon presentation of pneumonia in patients with and without sepsis. Interestingly, very old with sepsis presented less pleuritic pain and more altered mental status than patients without sepsis. These findings confirm the results of previous studies, which suggested that in elderly patients CAP may present without fever, cough or pleuritic pain but with altered mental status, a sudden decline in functional capacity, and worsening of previous comorbidities [2,35]. Physicians should be alert in the diagnosis of pneumonia in elderly patients in order to reduce complications associated with a delay in the start of the empiric antimicrobial therapy.

Of note, a Greek study described four comorbidities that contribute to the development of sepsis in patients with CAP, namely diabetes mellitus, chronic heart failure, coronary heart disease, and dementia [36]. Our results supported the finding for diabetes mellitus but added chronic renal disease as a risk factor for sepsis. In addition, male sex was a risk factor; men generally have more chronic comorbidities than women, and these may affect the integrity of host defenses to infection, thereby increasing morbidity and mortality rates. Men also exhibit a higher prevalence of CAP [37,38].

Our data indicated that prior antibiotic therapy was protective against sepsis in very old patients with CAP, supporting the findings of two previous studies in patients with CAP [9,39]. One study indicated that previous antibiotic therapy was associated with a lower prevalence of septic shock at hospital admission and a lower need for invasive ventilation [39]. The other study simply showed that prior antibiotic therapy was a protective factor in patients with CAP and sepsis [9]. These results indicate that early and appropriate antimicrobial treatment may be crucial in avoiding sepsis in very old patients with CAP. Of course, unmeasured confounders (e.g., patients with better family care are generally treated earlier, etc.) should always be considered. Moreover, dementia and other comorbidities may mask the clinical presentation of an infection, thus delaying antimicrobial administration [40].

Although there was no significant difference in 30-day and ICU mortality rates between the two groups, patients with sepsis had higher in-hospital and 1-year mortality rates. This is in accordance with several studies reporting the burden of sepsis in the elderly population [6,9,11,38,41,42]. The mortality rate due to sepsis in the elderly has been reported to be 1.3–1.5 times higher than that in younger patients [43,44]. Together, these data highlight the need for prompt diagnosis and treatment in very old patients who develop sepsis.

In the multivariate analysis, chronic renal and neurological diseases were associated with an increased 30-day mortality in patients with CAP-related sepsis. This supports previous findings that the population-based prevalence of sepsis increases exponentially with age and number of comorbidities, and this, in turn, increases mortality risk. Diabetes mellitus was the only factor associated with a lower risk for 30-day mortality in patients with sepsis. The impact of diabetes mellitus on outcomes in patients with sepsis remains controversial. In a recent prospective cohort study from the Netherlands, mortality was compared in 241 diabetic and 863 non-diabetic patients with sepsis; no significant differences were found in short or long-term outcomes, inflammatory biomarker levels, coagulation factors, or endothelial activation [45]. By contrast, a study of 1.5 million critically ill patients suggested that diabetes may have a protective effect [46]. This may relate to higher tolerance of sustained levels of moderate hyperglycemia and greater adaptability to marked fluctuations in glycemia. Thus, patients without diabetes may be worse off for having a compromised immune response and an altered microvasculature, increasing the possibility of multiple organ dysfunction [47].

In our opinion, the most important strengths of this study are the large sample size, the prospective and consecutive data collection, and the focus on sepsis in very old patients with CAP. The exclusion of patients with DNR orders was dictated by the previous finding that this can be an important confounder in patients with sepsis [30,48]. There are, however, some limitations that need to be addressed. The prolonged recruitment period may have affected the results because patient care has evolved during this time. That said, the protocol for CAP management at our hospital has not changed substantially during this period. Second, we did not include data about dysphagia and aspiration pneumonia, which can be common in the very old. Third, data about the quality of life, frailty, and/or functional status were not recorded. Finally, because data were collected from a single academic teaching hospital in Spain, it may not be possible to extrapolate results to patients admitted to different hospitals or to other countries.

Unlike previous reports, which defined sepsis according to the Sepsis-2 criteria, this is the first study investigating sepsis in very old CAP patients according to its last definition (Sepsis-3 criteria). Due to the high prevalence of sepsis worldwide, it is of pivotal importance to understand its global epidemiology especially in very old patients and to guide clinicians in the early recognition of patients at risk of developing it. Of course, prompt antibiotic therapy is required to improve outcomes. Moreover, our study suggests the potential benefit of previous antibiotic therapy to protect against sepsis: in very old patients with a clear suspicion of CAP, early initiation of antibiotics in the setting of primary cares should be encouraged.

## 5. Conclusions

In conclusion, in very old patients with CAP who developed sepsis, both in-hospital and 1-year mortality are higher compared with their peers who have no sepsis. Antibiotic therapy given before admission was associated with a decreased risk of sepsis though this may be influenced by unmeasured confounders.

## Figures and Tables

**Figure 1 jcm-08-00961-f001:**
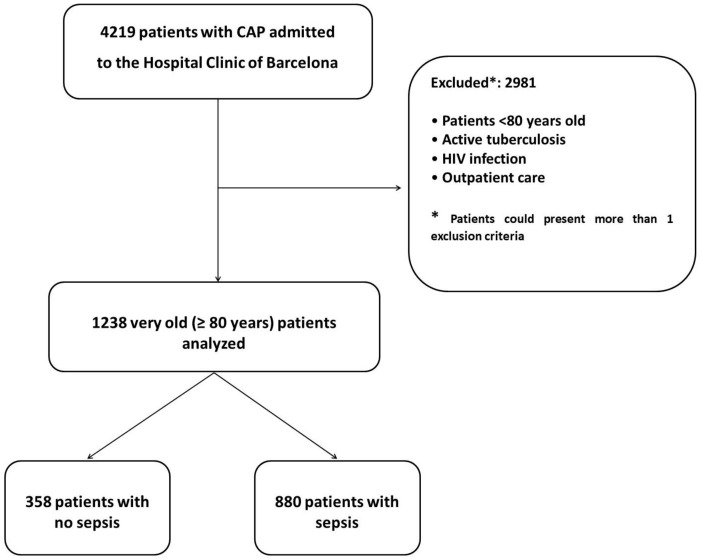
Flow chart of the study population.

**Table 1 jcm-08-00961-t001:** Characteristics of very old (≥80 years) patients by the presence or absence of sepsis.

	Sepsis		
Variable	No (*N* = 358)	Yes (*N* = 880)	*P*-Value
Age, mean (SD), years	86.6 (5.0)	85.9 (4.5)	0.030
Male sex, *n* (%)	155 (43)	531 (60)	<0.001
Current smoker, *n* (%)	10 (3)	45 (5)	0.081
Current alcohol user, *n* (%)	19 (6)	57 (7)	0.47
Previous antibiotic in last week, *n* (%)	98 (30)	190 (23)	0.020
Influenza vaccine, *n* (%)	184 (63)	468 (64)	0.89
Pneumococcal vaccine, *n* (%)	72 (25)	185 (25)	>0.99
Previous inhaled corticosteroids, *n* (%)	62 (18)	185 (22)	0.19
Previous systemic corticosteroids, *n* (%)	12 (3)	43 (5)	0.24
Fever, *n* (%)	231 (69)	569 (66)	0.72
Pleuritic pain, *n* (%)	91 (27)	177 (21)	0.021
Dyspnea, *n* (%)	261 (76)	662 (77)	0.78
Cough, *n* (%)	236 (69)	637 (74)	0.056
Altered mental status, *n* (%)	64 (19)	295 (34)	0.001
Prior pneumonia (last year), *n* (%)	33 (10)	131 (16)	0.011
Comorbidities, *n* (%) ^a^	275 (78)	710 (81)	0.28
Chronic respiratory disease	127 (38)	344 (40)	0.42
Chronic cardiovascular disease	67 (19)	191 (22)	0.28
Diabetes mellitus	70 (20)	241 (28)	0.007
Neurologic chronic disease	123 (36)	266 (31)	0.085
Chronic renal disease	22 (6)	113 (13)	0.001
Chronic liver disease	12 (3)	25 (3)	0.58
Nursing home resident, *n* (%)	71 (20)	156 (18)	0.36
Dyspnea, *n* (%)	261 (76)	662 (77)	0.78
Pleuritic pain, *n* (%)	91 (27)	177 (21)	0.021
Respiratory rate, median (IQR), breaths/min	24 (20; 30)	25 (22; 32)	0.017
C-reactive protein, median (IQR), mg/dL	16.2 (8.5; 24.4)	16.2 (7.4; 25.9)	0.95
Lymphocytes, median (IQR), cell/mm^3^	1002 (621; 1560)	888 (510; 1316)	0.002
Neuthophils, median (IQR), cell/mm^3^	9840 (6708; 14,130)	10,218 (6596;15,136)	0.37
PSI score, median (IQR)	107 (91; 127)	123.5 (104; 145)	<0.001
PSI risk class IV–V, *n* (%) ^b^	77 (78)	507 (92)	<0.001
Severe CAP, *n* (%)	21 (11)	288 (45)	<0.001
Bacteremia, *n* (%) ^c^	19 (9)	79 (13)	0.086
Pleural effusion, *n* (%)	42 (13)	110 (13)	0.90
Multilobar appearance on CXR, *n* (%)	74 (21)	203 (23)	0.36
Septic shock at admission, *n* (%)	0 (0)	61 (7)	<0.001
Do-not-resuscitate order, *n* (%)	59 (18)	148 (18)	0.96
Empiric antibiotic therapy, *n* (%)	-	-	-
Monotherapy	145 (41)	310 (36)	0.072
Fluoroquinolones	91 (26)	187 (21)	0.10
β-lactams	53 (15)	120 (14)	0.57
Other therapy	1 (0.3)	3 (0.3)	>0.99
Combination therapies	208 (59)	561 (64)	0.072
β-lactams plus macrolides	105 (30)	255 (29)	0.87
β-lactams plus fluoroquinolones	70 (20)	227 (26)	0.021
Other combination therapies	33 (9)	79 (9)	0.88
Appropriate empiric treatment, *n* (%)	339 (97)	817 (96)	0.57

Percentages calculated with non-missing data only. ^a^ May have >1 comorbid condition. ^b^ Stratified according to 30-day mortality risk for CAP: classes I–III (≤90 points) had low mortality risk while classes IV–V (>90 points) had the highest mortality risk. ^c^ Calculated only for patients with blood samples (215 patients in the no sepsis group and 594 patients in the sepsis group to calculate the percentages. Abbreviations: CAP, community-acquired pneumonia; IQR, interquartile range; PSI, Pneumonia Severity Index; SD, standard deviation; IQR inter-quartile range.

**Table 2 jcm-08-00961-t002:** Significant risk factors for sepsis in the logistic regression analyses (*n* = 1238).

	Univariate ^a^			Multivariable ^b,c^		
Variable	OR	95% CI	*P*-Value	OR	95% CI	*P*-Value
Male sex	1.97	1.54 to 2.53	<0.001	1.87	1.46 to 2.41	<0.001
Smoking status ^d^	-	-	<0.001	-	-	-
Never smoker	1.00	-	-	-	-	-
Current smoker	2.00	1.02 to 3.94	0.044	-	-	-
Ex-smoker	1.58	1.21 to 2.06	0.001	-	-	-
Previous antibiotic in last week	0.70	0.53 to 0.92	0.010	0.71	0.54 to 0.94	0.016
Chronic renal disease	2.30	1.43 to 3.69	0.001	2.08	1.29 to 3.37	0.003
Diabetes mellitus	1.52	1.13 to 2.04	0.006	1.42	1.05 to 1.92	0.024
Neurological chronic disease	0.79	0.61 to 1.02	0.074	-	-	-

Data are shown as estimated ORs (95% CIs) of the explanatory variables in the sepsis group. OR is defined as the probability of being in the sepsis group divided by the probability of being in the no-sepsis group. The *P*-values are based on the null hypothesis that all ORs relating to an explanatory variable equal unity (no effect). ^a^ The variables analyzed in the univariate analysis were as follows: gender, smoking status, alcohol consumption, influenza vaccine, pneumococcal vaccine, previous inhaled corticosteroids, previous systemic corticosteroids, previous antibiotic in last week, chronic pulmonary disease, chronic cardiovascular disease, chronic renal disease, chronic liver disease, diabetes mellitus, chronic neurologic disease, and nursing home residency. ^b^ Hosmer–Lemeshow goodness-of-fit test, *p* = 0.86. ^c^ Predictors from the model can be used to calculate the probability of sepsis by the following formula: Exp(β)/(1 + Exp(β)), where β = 0.514 + 0.628 (in case of male) − 0.344 (in case of previous antibiotic) + 0.734 (in case of acute renal failure) + 0.349 (in case of diabetes mellitus). Using this model, the probability of sepsis for patients without any of three risk factors and with the protective factor was 54%, and 90% for patients showing all three risk factors and without the protective factor. ^d^ The *p*-value corresponds to differences between the three groups (never smoker, current smoker, or ex-smoker). Abbreviations: CI, confidence interval; OR, odds ratio.

**Table 3 jcm-08-00961-t003:** Clinical outcomes according to sepsis.

	Sepsis		
Variable	No (*N* = 358)	Yes (*N* = 880)	*P*-Value
Length of hospital stay, median (IQR), days	7 (6; 11)	8 (6; 13)	0.005
In-hospital mortality, *n* (%)	32 (9)	129 (15)	0.006
30-day mortality, *n* (%)	40 (11)	134 (15)	0.062
One-year mortality, *n* (%)	56 (16)	190 (22)	0.013
ICU admission, *n* (%)	17 (5)	119 (14)	<0.001
ICU mortality, *n* (%) ^a^	1 (6)	20 (17)	0.47
Mechanical ventilation, *n* (%) ^b^	-	-	<0.001
Not ventilated	237 (99)	582 (89)	<0.001
Non-invasive	1 (0.4)	38 (6)	<0.001
Invasive	2 (1)	34 (5)	0.003

^a^ Calculated only for patients admitted to the ICU (17 patients in the no sepsis group and 119 patients in the sepsis group were used to calculate the percentages). ^b^ Patients who received initially non-invasive ventilation but needed subsequently intubation were included in the invasive mechanical ventilation group. Abbreviations: ICU, intensive care unit; IQR, interquartile range.

**Table 4 jcm-08-00961-t004:** Significant risk factors for 30-day mortality in patients with sepsis in weighted logistic regression analyses (*n* = 640).

	Univariate ^a^			Multivariable ^b^		
Variable	OR	95% CI	*P*-Value	OR	95% CI	*P*-Value
Chronic renal disease	2.60	1.40 to 4.83	0.003	2.57	1.35 to 4.89	0.004
Diabetes mellitus	0.49	0.22 to 0.88	0.021	0.44	0.21 to 0.89	0.023
Neurologic chronic disease	2.68	1.60 to 4.50	<0.001	2.80	1.62 to 4.85	<0.001
Nursing home	2.19	1.21 to 3.94	<0.001	-	-	-
PSI risk class IV–V	4.09	0.98 to 17.0	0.054	-	-	-

Data are shown as estimated ORs (95% CIs) of the explanatory variables in the 30-day mortality group. OR is defined as the probability of membership of the group 30-day mortality divided by the probability of membership of the non-30-day mortality group. The *P*-value is based on the null hypothesis that all ORs relating to an explanatory variable equal unity (no effect). ^a^ The variables analyzed in the univariate analysis were: gender, smoking status, alcohol consumption, influenza vaccine, pneumococcal vaccine, previous inhaled corticosteroids, previous systemic corticosteroids, previous antibiotic in last week, chronic pulmonary disease, chronic cardiovascular disease, chronic renal disease, chronic liver disease, diabetes mellitus, neurological disease, nursing home residency, PSI risk class, C-reactive protein, lymphocytes, empiric antibiotic treatment, and appropriate empiric antibiotic treatment. ^b^ Summary statistics of the multivariable model: Pearson chi-square test, value/df = 0.82; AIC value = 163.06. Abbreviations: AIC, Akaike’s Information Criterion; CI, confidence interval; OR, odds ratio; PSI, Pneumonia Severity Index.

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
