# Peer review of "Risk and Prognostic Factors in Very Old Patients with Sepsis Secondary to Community-Acquired Pneumonia"

_jcm, 2019, doi:10.3390/jcm8070961_

Round 1
Reviewer 1 Report
This paper is interesting ,before this draft for publication
I have the comment as below
1) the authors can give us the data such as the how many patients received the thoracic imaging such as the chest -x-ray , the sputum culture and smear, blood culture?
2) how many patients have diagnosis as the pulmonary tuberculosis, did author exclude the active pulmonary tuberculosis from the study group?
Author Response
Reviewer 1: This paper is interesting, before this draft for publication I have the comment as below
1) the authors can give us the data such as the how many patients received the thoracic imaging such as the chest -x-ray , the sputum culture and smear, blood culture?
Answer: Thanks for the comment. All patients included in the study received chest-x-ray (n=1,238). Sputum cultures were performed in 436 (35.2%) patients, nasopharyngeal swabs in 427 (34.4%) and blood cultures in 809 (65.3%). We included these data in the supplementary material, page 4, result section.
2) how many patients have diagnosis as the pulmonary tuberculosis, did author exclude the active pulmonary tuberculosis from the study group?
Answer: Thanks for the comment. Sixteen cases of pulmonary tuberculosis were diagnosed during the study period and all were excluded from the analysis as we described in methods (study design and patients) page 5, line 89.

Reviewer 2 Report
This manuscript is, " Risk and Prognostic Factors in Very Old Patients with Sepsis Secondary to Community-Acquired Pneumonia?" by Dr. Catia Cillóniz and his colleagues. This study is a one-center, retrospective, long-term cohort study dealing with the elderly having CAP to demonstrate the predictors of sepsis and 30-day mortality. The manuscript presents the results of analyses of patent demography and clinical outcome data. The manuscript is well writing and fully constructive. In my opinion, there were several problematic concerns as the following
First, the term of sepsis adapted in your work is my leading concern. Because of the lacking association of sepsis presentation and 30-day mortality, it was confused that the coexistence of sepsis in CAP patients was regarded as the one of outcome parameters in the study design. In generally, the atypical presentations of CAP in the elderly was documented in the literature. In my opinion, it is useful information to remind clinicians to diagnose CAP among the elderly having atypical presentations or lacking sepsis; as well as to demonstrate the clinical predictors of non-sepsis presentations among the elderly having CAP.
Second, in this work, numerous predictors of 30-day mortality in CAP patients with sepsis was identified, particularly for pneumonia and comorbidity severity. However, these findings were not enough novel.
Third, the empirical antimicrobial class was analyzed in your work (Table 1). But, the effect of inappropriate empirical therapy on patient outcomes was lacking in the manuscript.
Fourth, for the elderly, the covariate of comorbidity severity (such as McCabe-Johnson classification and Charlson severity index score) and daily performance (ADL) should be enrolled in analyses because the crucial association of daily performance (ADL) and patient prognoses was well established.
Author Response
First, the term of sepsis adapted in your work is my leading concern. Because of the lacking association of sepsis presentation and 30-day mortality, it was confused that the coexistence of sepsis in CAP patients was regarded as the one of outcome parameters in the study design. In generally, the atypical presentations of CAP in the elderly was documented in the literature. In my opinion, it is useful information to remind clinicians to diagnose CAP among the elderly having atypical presentations or lacking sepsis; as well as to demonstrate the clinical predictors of non-sepsis presentations among the elderly having CAP.
Answer: Thanks for the comment. In our study we used the definition of Sepsis according to the SEPSIS -3 criteria[1].
Although we did not find any association between sepsis presentation and 30-day mortality, the comparison between the sepsis and no-sepsis group in terms of 30-day mortality was at the limit of statistical significance (11% vs. 15%, p 0.062).
In our study we observed that patients with and without sepsis may have an uncommon presentation of pneumonia: We included these data in table 1, in the paragraph Results (page 6, lines 183-186) and in the paragraph Discussion (page 9, lines 274- 280).
Clinical presentation | No Sepsis (n= 358) | Sepsis (n=880) | P value |
Fever, n (%) | 231 (69) | 569 (66) | 0.72 |
Pleuritic pain, n (%) | 91 (27) | 177 (21) | 0.021 |
Dyspnea, n (%) | 261 (76) | 662 (77) | 0.78 |
Cough, n (%) | 236 (69) | 637 (74) | 0.056 |
Altered mental status, n (%) | 64 (19) | 295 (34) | 0.001 |
Moreover, in the supplementary material (Results section, page 5 and table 1) we described the differences in the clinical presentation of pneumonia between patients aged 65-79 years and ≥80 years: “The uncommon presentation of pneumonia was frequent in very old patients: compared with patients aged 65-79 years, they presented a lower rate of fever, cough, pleuritic pain and a higher frequency of altered mental status at admission”. “Also, this uncommon presentation of pneumonia was observed in the subgroup of patient with and without sepsis”.
Clinical presentation | 65- 79 y (n= 975) | ≥80 y (n=1238) | P value |
Fever, n (%) | 717 (75) | 800 (66) | <0.001< span=""> |
Pleuritic pain, n (%) | 325 (34) | 268 (23) | <0.001< span=""> |
Dyspnea, n (%) | 709 (74) | 923 (77) | 0.13 |
Cough, n (%) | 730 (76) | 873 (73) | 0.058 |
Altered mental status, n (%) | 216 (22) | 359 (30) | <0.001< span=""> |
“Also, this uncommon presentation of pneumonia was observed in the subgroup of patient with and without sepsis”.
Clinical presentation | Sepsis 65- 79 y (n= 703) | Sepsis ≥80 y (n=880) | P value |
Fever, n (%) | 503 (73) | 569 (66) | 0.002 |
Pleuritic pain, n (%) | 215 (31) | 177 (21) | <0.001< span=""> |
Dyspnea, n (%) | 525 (76) | 662 (77) | 0.58 |
Cough, n (%) | 529 (77) | 637 (74) | 0.27 |
Altered mental status, n (%) | 190 (27) | 295 (34) | 0.003 |
Clinical presentation | NO-Sepsis 65- 79 y (n= 272) | NO- Sepsis ≥80 y (n=358) | P value |
Fever, n (%) | 214 (80) | 231 (67) | <0.001< span=""> |
Pleuritic pain, n (%) | 110 (42) | 91 (27) | <0.001< span=""> |
Dyspnea, n (%) | 184 (69) | 261 (76) | 0.057 |
Cough, n (%) | 201 (75) | 236 (69) | 0.078 |
Altered mental status, n (%) | 26 (10) | 64 (19) | 0.002 |
Second, in this work, numerous predictors of 30-day mortality in CAP patients with sepsis was identified, particularly for pneumonia and comorbidity severity. However, these findings were not enough novel.
Answer: Thanks for the comment. To date, information regarding sepsis in very old (≥80 years old) patients with community-acquired pneumonia (CAP), using the SEPSIS-3 definition, is limited. This is the first study addressing risk and prognostic factors of sepsis based on the SEPSIS-3 definition in very old patients with CAP. We recognize that findings are not striking but they are important for the reasons mentioned.
We agree with the reviewer that numerous predictors of 30-day mortality in CAP patients with sepsis have been identified yet; however, our study was focused on a specific subgroup of patients, that is very old patients with CAP. In our analysis, chronic renal disease, diabetes mellitus and male sex were independent risk factors for sepsis, whereas antibiotic therapy in the previous week was a protective factor. We believe that these findings should be considered when visiting this subgroup of patients, even outside the hospital.
Third, the empirical antimicrobial class was analyzed in your work (Table 1). But, the effect of inappropriate empirical therapy on patient outcomes was lacking in the manuscript.
Answer: Thanks for the comment. The variables included in the multivariable analysis are described in a footnote of table 4. We included appropriate empiric antibiotic treatment in the analysis; however we did not find any association between this variable and 30-day mortality (OR 0.80 95%CI 0.23 to 2.75, p: 0.72).
Fourth, for the elderly, the covariate of comorbidity severity (such as McCabe-Johnson classification and Charlson severity index score) and daily performance (ADL) should be enrolled in analyses because the crucial association of daily performance (ADL) and patient prognoses was well established.
Answer: Thanks for the comment. Unfortunately, variables about quality of life, frailty, and/or functional status were not recorded in our data base. We included this limitation in the new version of the article, page 10, line 316- 317.

Round 2
Reviewer 2 Report
Thanks for your substantial inputs according to my concerns and opinions. Indeed, I agree that that there were numerous atypical symptoms/signs at initial pneumonia presentations in the elderly. However, in the literature, there were also several established reports dealing with these issues, such as the Journal of the American Geriatrics Society 1989;37:867-872 (Cynthia Harper, et al.) and Journal of the American Geriatrics Society 2000;48:1316-1320 (Jerry C. Johnson, et al.)
Accordingly, your work was not novel enough.
Author Response
Thanks for the comment. We included a paragraph in the discussion section about the importance and novelty of our article. Page 10, line 328 – 335.